# Expression and immunological role of FUNDC2 in pan-cancer

Xirong Qiu *, Shuyu Wang, Chenlu Li, Yinan Wang

Department of Pharmacology, School of Medicine, Lijiang Culture and Tourism College, Lijiang, Yunnan, China

* ynkmq6688@outlook.com

## Abstract

FUNDC2 is a novel mitochondrial protein and is highly involved in various cancers. However, expression pattern and possible role and mechanism of FUNDC2 in pan-cancer remain to be investigated. TIMER 2.0 was used to investigate the expression patterns and immune infiltration of *FUNDC2*. GEPIA was applied to study the relationship between level of *FUNDC2* and prognosis of the patients with pan-cancer. STRING was employed to analyze the potential interacting proteins of FUNDC2. The phosphorylation sites were predicted by cBioPortal and PhosphoNet. Furthermore, variations of FUNDC2 in cancers were investigated by cBioPortal. Finally, AlphaFold was used to predict the structure of FUNDC2. The data show that there were significant differences in the expression levels of *FUNDC2* between cancer tissues and controls. Specifically, the levels of *FUNDC2* in 8 cancers were significantly lower than the respective controls. The survival time of the cancer patients with higher levels of *FUNDC2* was longer than that of lower *FUNDC2* in most different types of cancers. The pattern of FUNDC2 was significantly related to immune infiltration of B cells of cancer patients. STRING analysis revealed that FUNDC2 can interact with FUNDC1, et al. Fifteen phosphorylation sites were predicted by PhosphoNet and cBioPortal, of which the S167 also overlapped with the mutation sites of FUNDC2. These data collectively show that the mitochondrial protein FUNDC2 may serve as a possible prognostic biomarker across various cancers and the mechanism may include immune infiltration.

## Introduction

FUN14 domain-containing protein 2 (FUNDC2) with 176 amino acids is a novel mitochondrial protein [1]. It is also involved in cell apoptosis and ferroptosis, which is highly involved in various cancers [2–5]. However, expression level and clinical significance of FUNDC2 in cancers remain to be investigated.

**Data availability statement:** All relevant data can be obtained from the TIMER 2.0 database (http://cistrome.org/TIMER/), GEPIA (http://gepia.cancer-pku.cn/), STRING database (https://string-db.org/), cBioportal database (http://www.cbioportal.org), Phosphonet website (http://www.phosphonet.ca/default.aspx), AlphaFold (https://alphafold.ebi.ac.uk/), Uniprot database (https://www.uniprot.org/).

**Funding:** The author(s) received no specific funding for this work.

**Competing interests:** All authors declare that they have no conflict of interest.

FUNDC2 participated in triple-negative breast cancer (TNBC), and hepatocarcinoma [6–8]. Specifically, depletion of *FUNDC2* inhibited cell proliferation, migration, and invasion in TNBC [6]. The expression level of FUNDC2 in luminal breast cancer tissues was higher than the controls. Notably, low expression of FUNDC2 was correlated to longer progression-free survival in TNBC patients [6]. FUNDC2 overexpression was inversely correlated with the survival of the patients with hepatocellular carcinoma (HCC) [7]. The expression of FUNDC2 in cancer tissues were lower than the normal tissues in most HCC samples. Additionally, FUNDC2 was also implicated in cervical cancer [8]. Notably, FUNDC2 served as a useful indicator for ductal carcinoma *in situ* (DCIS) staging and differentiated between DCIS and invasive breast cancer [9]. LASSO regression and univariate Cox analyses further validated the significance of FUNDC2 in cancer prognosis, emphasizing its potential as a diagnostic and prognostic marker. However, the possible role and mechanism of FUNDC2 in pan-cancer remains to be investigated.

TIMER 2.0, GEPIA, STRING, cBioPortal, and PhosphoNet databases were employed in this study to analyze the expression and prognostic role of FUNDC2 in pan-cancer, and the relationship between *FUNDC2* expression and the degree of immune infiltration was examined [10–13]. Collectively, the data demonstrate that *FUNDC2* expression is involved in the prognosis of cancer patients possibly through its immune infiltration of various immune cells.

## Materials and methods

### Expression of *FUNDC2* was investigated by using TIMER 2.0

The mRNA expression of *FUNDC2* was analyzed in various cancer types in the TIMER 2.0 database (http://cistrome.org/TIMER/) as described previously [13–14].

### Analyzing survival of cancer patients in GEPIA

The relationship between the expression of *FUNDC2* and survival in pan-cancer was analyzed by using GEPIA (http://gepia.cancer-pku.cn/). An online tool of GEPIA was selected to analyze the gene expression profiles based on RNA sequencing data by GTEx and the TCGA [15–17]. The effects of *FUNDC2* expression on OS and DFS in each of the 33 available cancer types (total number = 33) were examined. Hazard ratios (HRs) were calculated using 95% confidence intervals (CI) and log-rank *P*-values.

### The relationship between *FUNDC2* expression and immune infiltration was studied by applying TIMER 2.0

The relationship between *FUNDC2* expression and immune infiltration was examined using TIMER 2.0 (http://cistrome.org/TIMER/). The Cancer Genome Atlas (TCGA) provided a comprehensive analysis of immune infiltration in multiple cancer types using six state-of-the-art algorithms and four modules. This resource was an excellent resource for evaluating the molecular characterization of tumor immune infiltration relationships. The expression of *FUNDC2* was examined in all five types of immune cells present in immune infiltration cells including B cells. Furthermore,

the expression level of *FUNDC2* correlated with tumor purity. Specifically, the "gene module" in the TIMER 2.0 (http://cistrome.org/TIMER/) allows us to select "*FUNDC2*" and visualize the possible correlation of the expression of *FUNDC2* with the immune infiltration pattern of B cells, et al. across various cancer types in this study. Once *FUNDC2* and immune infiltrates of the B cells, et al. are submitted, a heatmap will illustrate the purity-adjusted Spearman's rho in diverse cancer types.

### Network of protein interactions with FUNDC2

A PPI (protein-protein interaction) network was created using the STRING database (https://string-db.org/) with *Homo sapiens* as the defined species [18–30]. The interaction proteins of FUNDC2 were demonstrated.

### Variations of FUNDC2 in pan-cancer

The cBioportal database (http://www.cbioportal.org), which contains somatic mutations, Copy number variants (CNVs), and cancer genomics information from the TCGA and GEO databases was used in this study. The single nucleotide mutations of *FUNDC2* in the pan-cancer cohort and mutation types were investigated by using the cBioPortal [31–35].

### Analyses of the phosphorylation sites of FUNDC2

The PhosphoNet website (http://www.phosphonet.ca/default.aspx) provides a comprehensive compilation of over 200000 documented phosphorylation sites. The majority of protein phosphorylation occurs at serine (Ser or S), threonine (Thr or T), and tyrosine (Tyr or Y) residues [36]. The phosphorylation sites of FUNDC2 were predicted by the PhosphoNet.

### Pathogenic probability analysis of FUNDC2

AlphaFold (https://alphafold.ebi.ac.uk/) is an important tool accurately predicting the atomic structure of any protein [37–40]. AlphaFold was utilized in this study to forecast the probability of pathogenesis after mutation of the FUNDC2. Because the Universal Protein (UniProt) database (https://www.uniprot.org/) is the most extensive repository of protein data, the transmembrane domain of FUNDC2 and the correlation between mutation locations and pathogenesis were investigated [41–50].

## Results

### mRNA expression level of *FUNDC2* in tumor tissues was lower than controls in most pan-cancers

Expression of FUNDC2 in pan-cancer was investigated by employing TIMER 2.0. The data demonstrated that there were significant differences between tumor tissues and their respective control tissues in 15 distinct cancers, of which the expression pattern of *FUNDC2* mRNA in 8 tumor tissues was lower than the controls in most pan-cancers, especially BLCA (Fig 1). Collectively, these results indicate that FUNDC2 mRNA in normal tissues is higher than their respective cancer tissues.

### Higher level of *FUNDC2* was significantly correlated with longer survival time of cancer patients

The possible relationship between *FUNDC2* mRNA expression and prognosis of the patients with various cancers was studied by using GEPIA. Fig 2 illustrated 7 cancer patients with higher expression levels of *FUNDC2* mRNA had a longer overall survival time (OS) or disease-free survival (RFS) in all the 16 different types of cancers.

### Pattern of FUNDC2 expression was statistically correlated to immune infiltration in immune cells of cancer patients

The possible association between FUNDC2 levels and immune infiltration of pan-cancer was explored by applying TIMER 2.0. The results in Fig 3 exhibited that the expression level of FUNDC2 were significantly and negatively correlated with

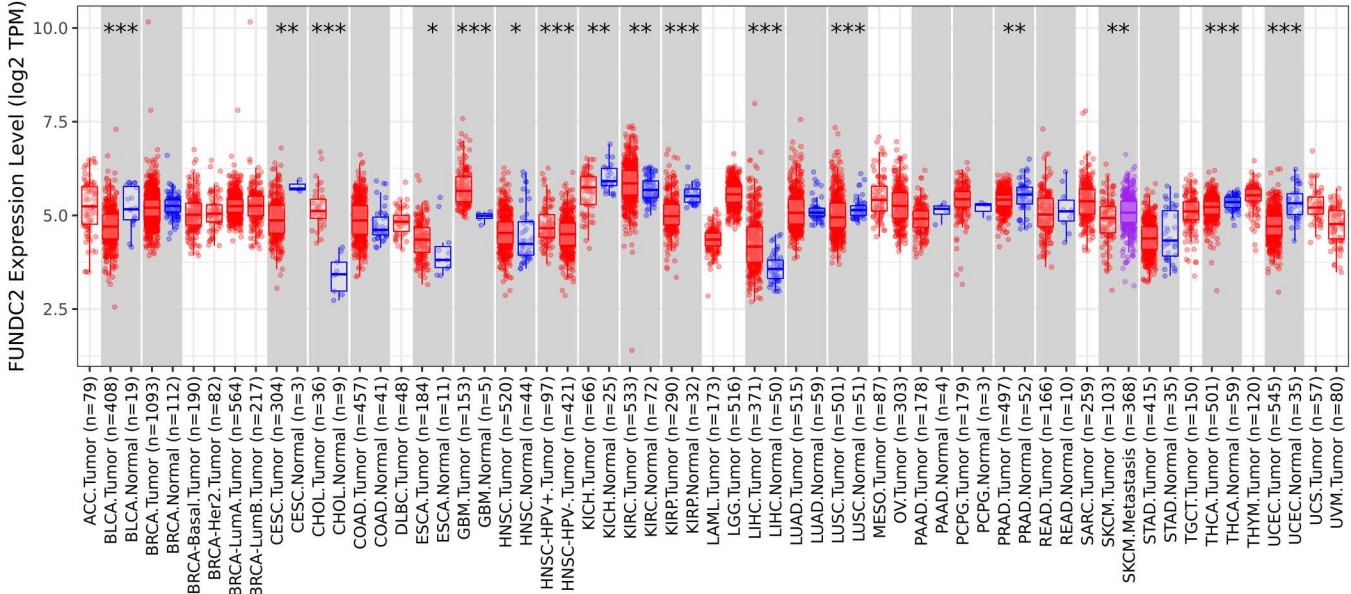

**Fig 1. Expression levels of *FUNDC2* in pan-cancer.** The expression levels of FUNDC2 mRNA in the cancer tissues and the respective controls of the patients with pan-cancer are analyzed by Timer 2.0 (http://timer.cistrome.org/). Distributions of gene expression levels are displayed The box plots are employed to display the distributions of the expression levels of the *FUNDC2* mRNA. *P < 0.05, **P <0.01, ***P <0.001. Adrenocortical carcinoma (ACC), Bladder urothelial carcinoma (BLCA), Breast invasive carcinoma (BRCA), Cervical squamous cell carcinoma and endocervical adenocarcinoma (CESC), Cholangiocarcinoma (CHOL), Colon adenocarcinoma (COAD), Lymphoid neoplasm diffuse large B-cell lymphoma (DLBC), Esophageal carcinoma (ESCA), Glioblastoma multiforme (GBM), Head and neck squamous cell carcinoma (HNSC), Kidney chromophobe (KICH), Kidney renal clear cell carcinoma (KIRC), Kidney renal papillary cell carcinoma (KIRP), Acute myeloid leukemia (LAML), Brain lower grade glioma (LGG), Liver hepatocellular carcinoma (LIHC), Lung adenocarcinoma (LUAD), Lung squamous cell carcinoma (LUSC), Mesothelioma (MESO), Pancreatic adenocarcinoma (PAAD), Pheochromocytoma and paraganglioma (PCPG), Prostate adenocarcinoma (PRAD), Rectum adenocarcinoma (READ), Ovarian serous cystadenocarcinoma (OV), Sarcoma (SARC), Skin cutaneous melanoma (SKCM), Stomach adenocarcinoma (STAD), Testicular germ cell tumors (TGCT), Thyroid carcinoma (THCA), Thymoma (THYM), Uterine corpus endometrial carcinoma (UCEC), Uterine carcinosarcoma (UCS), Uveal melanoma (UVM).

the immune infiltration in B cells for the ACC in EPIC. In comparison, there was a significant and positive relationship between FUNDC2 levels and immune infiltration in QUANTISEQ and CIBERSORT-ABS (Fig 3).

### Interaction analysis of FUNDC2 by STRING

The potential interacting proteins with FUNDC2 were analyzed by employing the STRING. The results in Fig 4 demonstrated that there were 10 proteins interacting with FUNDC2 in the analysis, including FUNDC1, which was a confirmed interaction partner in experimental investigation [13].

### 15 possible phosphorylation sites in FUNDC2 were predicted by PhosphoNet

Phosphorylation modifications of FUNDC2 played pivotal roles in cell functions such as cell death. To this end, potential phosphorylation sites in FUNDC2 were predicted by applying PhosphoNet, which was the famous online tool in identifying phosphorylation sites for almost all proteins. The data in Fig 5 displayed that there were 15 potential phosphorylation sites of amino acids in FUNDC2, namely S10, T16, S20, S31, S32, T37, S43, S74, S77, Y81, S151, T156, S167, T176, and S189.

### FUNDC2 variation analysis revealed overlapping of phosphorylation sites with hotspots of cancer patients

Tumorigenesis was significantly affected by genetic alterations and also phosphorylation sites. In order to conduct a thorough examination of the genetic modifications of FUNDC2, cBioportal was utilized to evaluate a total of 10,953 cancer

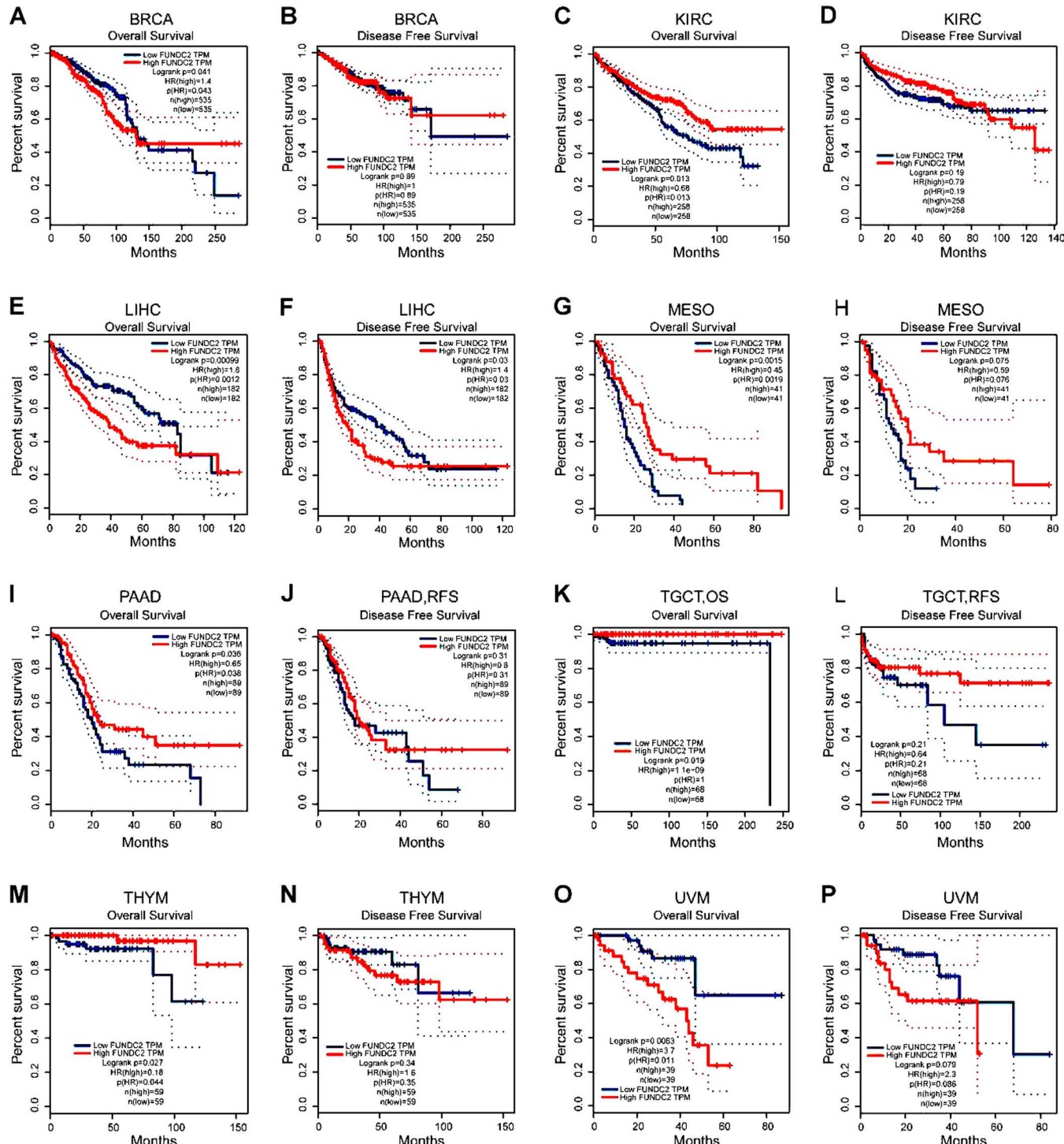

**Fig 2. High *FUNDC2* was significantly associated with long survival of the patients with pan-cancer generally.** The relationships between the expressions of FUNDC2 and the survival of the patients with diverse cancers (A-P) are studied by using GEPIA (http://gepia.cancer-pku.cn/). Both the overall survival (OS) and the disease-free survival (RFS) are exhibited respectively. *P* < 0.05 is regarded as significant in statistics. Bladder urothelial carcinoma (BLCA), Breast invasive carcinoma (BRCA), Kidney chromophobe (KICH), Kidney renal clear cell carcinoma (KIRC), Kidney renal papillary cell carcinoma (KIRP), Liver hepatocellular carcinoma (LIHC), Lung adenocarcinoma (LUAD), Lung squamous cell carcinoma (LUSC), Mesothelioma (MESO), Pancreatic adenocarcinoma (PAAD), Testicular germ cell tumors (TGCT), Thymoma (THYM), Uveal melanoma (UVM).

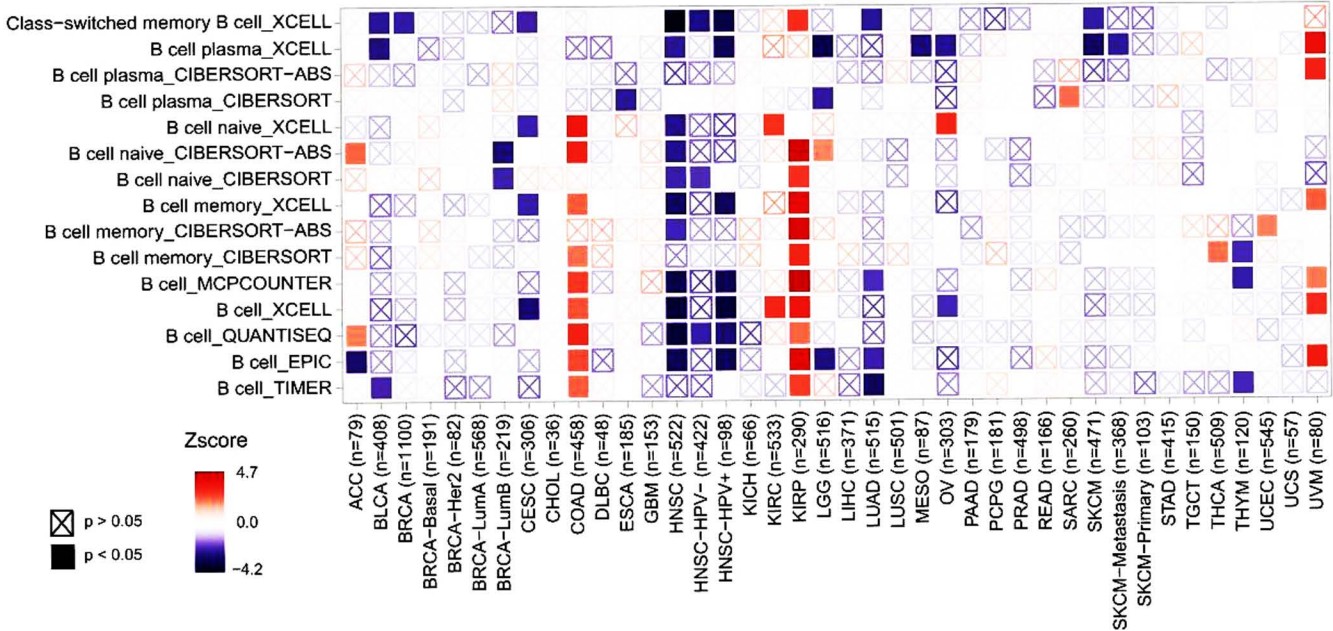

**Fig 3. Immune infiltration of B cells was correlated with *FUNDC2* expression in various cancers.** As a comprehensive resource for analyzing the immune infiltration of diverse immune cells including the B cells in this study, the TIMER (http://timer.cistrome.org/) is used across various cancer types systematically. *P* < 0.05 is regarded as significant in statistics.

patients and 10,967 samples. FUNDC2 mutation rates in cancer were relatively low (< 11%) (Fig 6A). 76 mutations in FUNDC2 were identified, including 30 missense mutations, 6 truncating mutations, 1 in-frame mutation, and 1 splice mutation (Fig 6B). The mutation rate of FUNDC2 was the highest in diffuse large B-cell lymphoma 10.42% (> 10%), followed by stomach adenocarcinoma (> 5%) (Fig 6A).

Both phosphorylation sites and cancer mutation hotspots in FUNDC2 were further analyzed to uncover the potential overlapping amino acids. Fig 4-6 illustrated clearly that a total of six sites were shared by both phosphorylation analysis by Phosphonet and hotspots of cancers, namely S10, S31, S151, S167, T176, and S189, respectively. Specifically, the 167 serine mutation in FUNDC2 was discovered in tumor patients.

Possible kinases have been predicted by employing the PhosphoNet online tool, of which the Ataxia Telangiectasia and Rad3-related protein (ATR, ATR serine/threonine-protein kinase) is most likely the protein kinase for FUNDC2 at Ser10, as shown in the S1 Fig. In addition, the phosphatases responsible for FUNDC2 may include PGAM family member 5, a mitochondrial serine/threonine protein phosphatase (PGAM5), Protein phosphatase-1 (PP1), PP2 or others [1–3,13,14]. Both FUNDC2 and PGAM5 are mitochondrial proteins, so it is reasonable to deduce that PGAM5 may be a potential phosphatase for FUNDC2. However, further studies should be done to validate the hypothesis.

## Pathogenicity probability of phosphorylation site mutations in FUNDC2

FUNDC2 was highly involved in various cancers especially breast cancers, phosphorylation modifications and structural changes also participate in anti-cancer therapy. Therefore, the structure of FUNDC2 and the pathogenicity of different sites in FUNDC2 were predicted by AlphaFold 2 (Fig 7A). The data exhibited that mutation of Serine 167 in FUNDC2 to Lysine (Lys or K) or Proline (Pro or P) showed pathogenicity rates of 0.587 and 0.747, respectively (Fig 7B), indicating that this site indeed plays a key role and also consistent with the clinical observations in cancer patients (Fig 6). UniProt

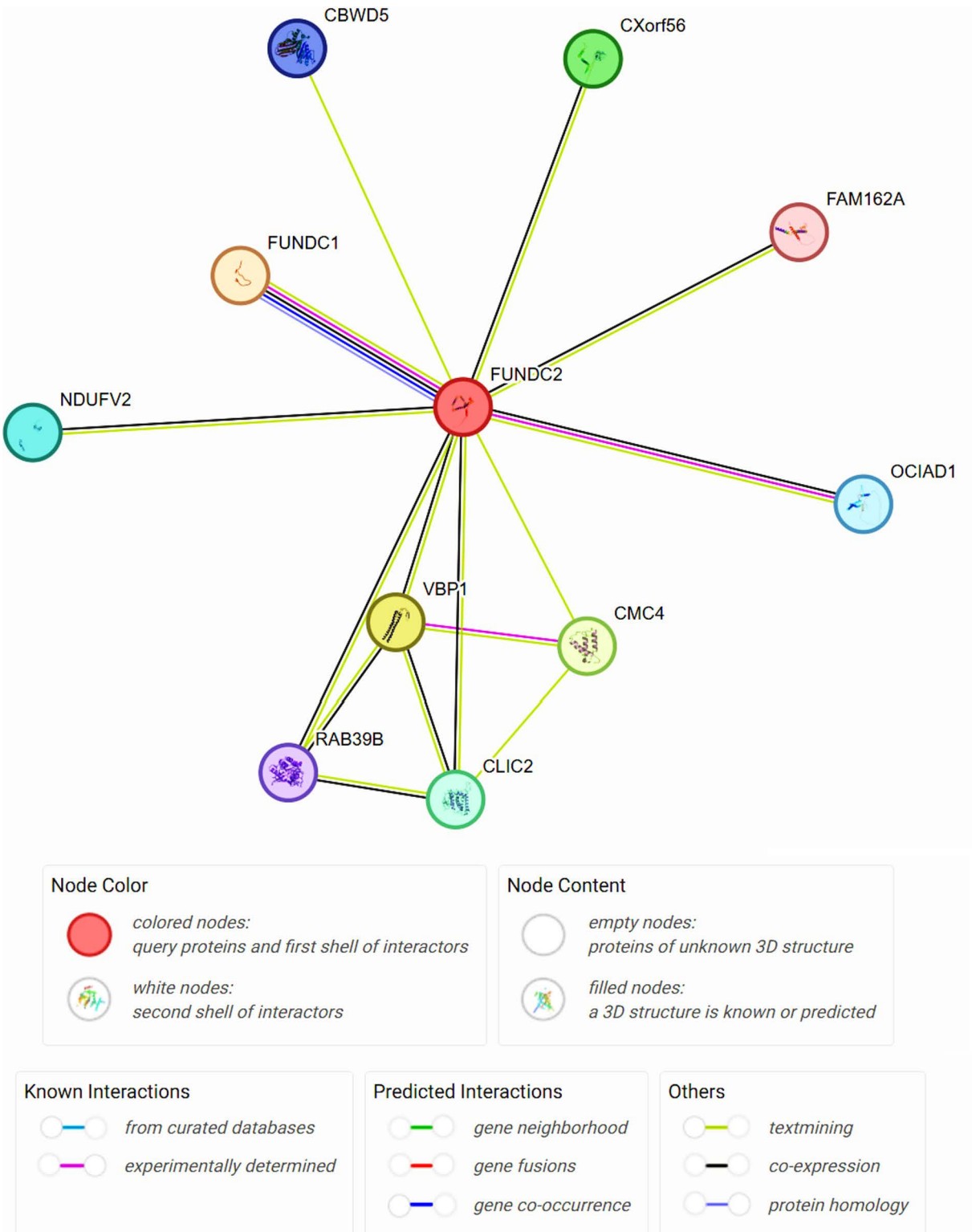

**Fig 4. The interaction proteins of FUNDC2 were predicted by using STRING.** The possible interacting partners of FUNDC2 are analyzed in STRING (https://cn.string-db.org/). The nodes in the above network represent the potential proteins interacting with FUNDC2.

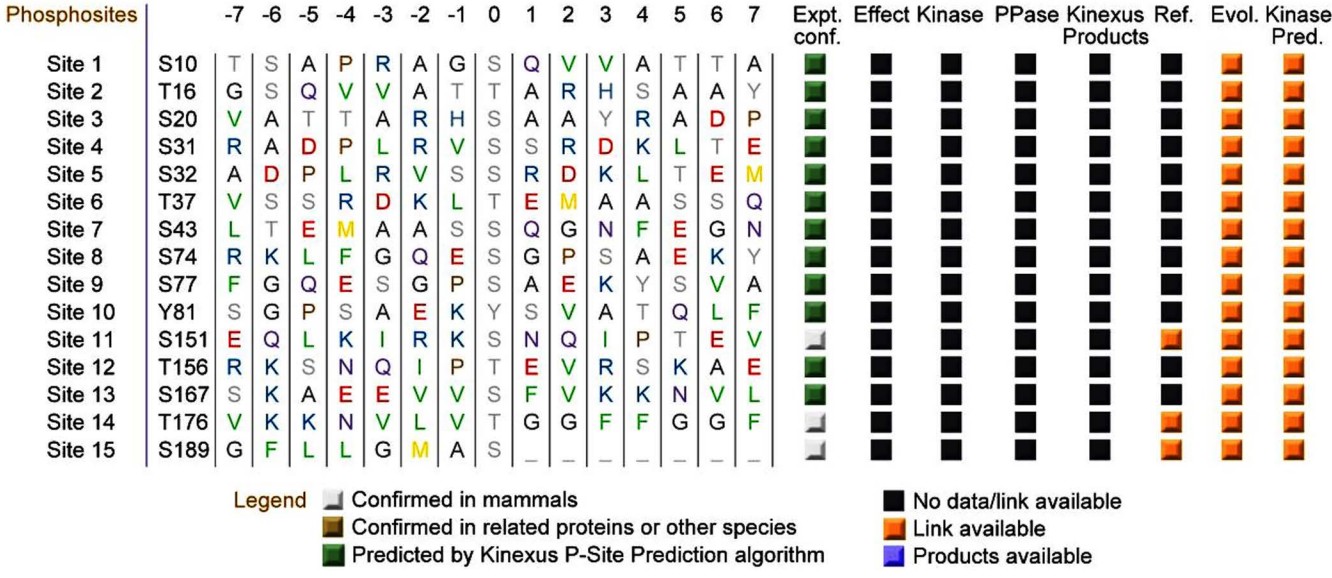

**Fig 5. Phosphorylation sites of FUNDC2 were predicted by PhosphoNet.** The possible phosphorylation sites of FUNDC2 are analyzed by applying PhosphoNet (http://www.phosphonet.ca/). Numerous parameters are selected as displayed in the methods in predicting the potential phosphorylation sites, and a score representing the probability is given. Specifically, the P-Sites confirmed by experiments usually have a lower hydrophobicity score.

results show that FUNDC2 contains three transmembrane domains and three topological domains (Fig 7C), in which both 167 S and 176 T were included in these structures.

## Discussion

The data in the current investigation show that the levels of FUNDC2 in cancer tissues were higher than those of controls generally. Furthermore, the survival time of the cancer patients with higher levels of FUNDC2 was longer than that of the lower patients. The expression of FUNDC2 was significantly correlated with immune infiltration of B cells and other immune cells in various cancers. Phosphorylation sites and variation of FUNDC2 overlapped, indicating that the S167L may be important in cancers. Moreover, FUNDC2 interacted with FUNDC1, et al. Collectively, these results suggested that FUNDC2 may serve as a potential biomarker in cancer diagnosis and prognosis.

The biggest novelty of this study is that FUNDC2 is involved in most different types of cancers. The level of FUNDC2 was significantly associated with the survival of cancer patients.

Previous investigations involving breast cancer, HCC and cervical carcinoma demonstrated that FUNDC2 participated in cancers, supporting our findings that FUNDC2 also was involved in pan-cancer [6]. Our analysis revealed that there is a higher FUNDC2 expression in breast cancer tissues compared to the non-tumor controls, based on a cohort of 140 breast cancer cases, 33 non-tumor benign lesions, 82 non-TNBC cases, and 58 TNBC cases. The present study further demonstrated that FUNDC2 expression was negatively correlated with OS and RFS in early-stage BRCA patients but positively correlated in later stages. Additionally, FUNDC2 overexpression in LIHC was detrimental to patient survival, consistent with previous TCGA data on LIHC transcription levels and Kaplan-Meier analysis [7]. Notably, while 54 LIHC samples showed little increase in FUNDC2 levels compared to 32 adjacent tissues, the reason may lie in different subjects [7].

There are some limitations in this study. Firstly, the relationship between FUNDC2 expression and patient outcomes across various cancer types should be addressed with more cases. Secondly, a correlation between

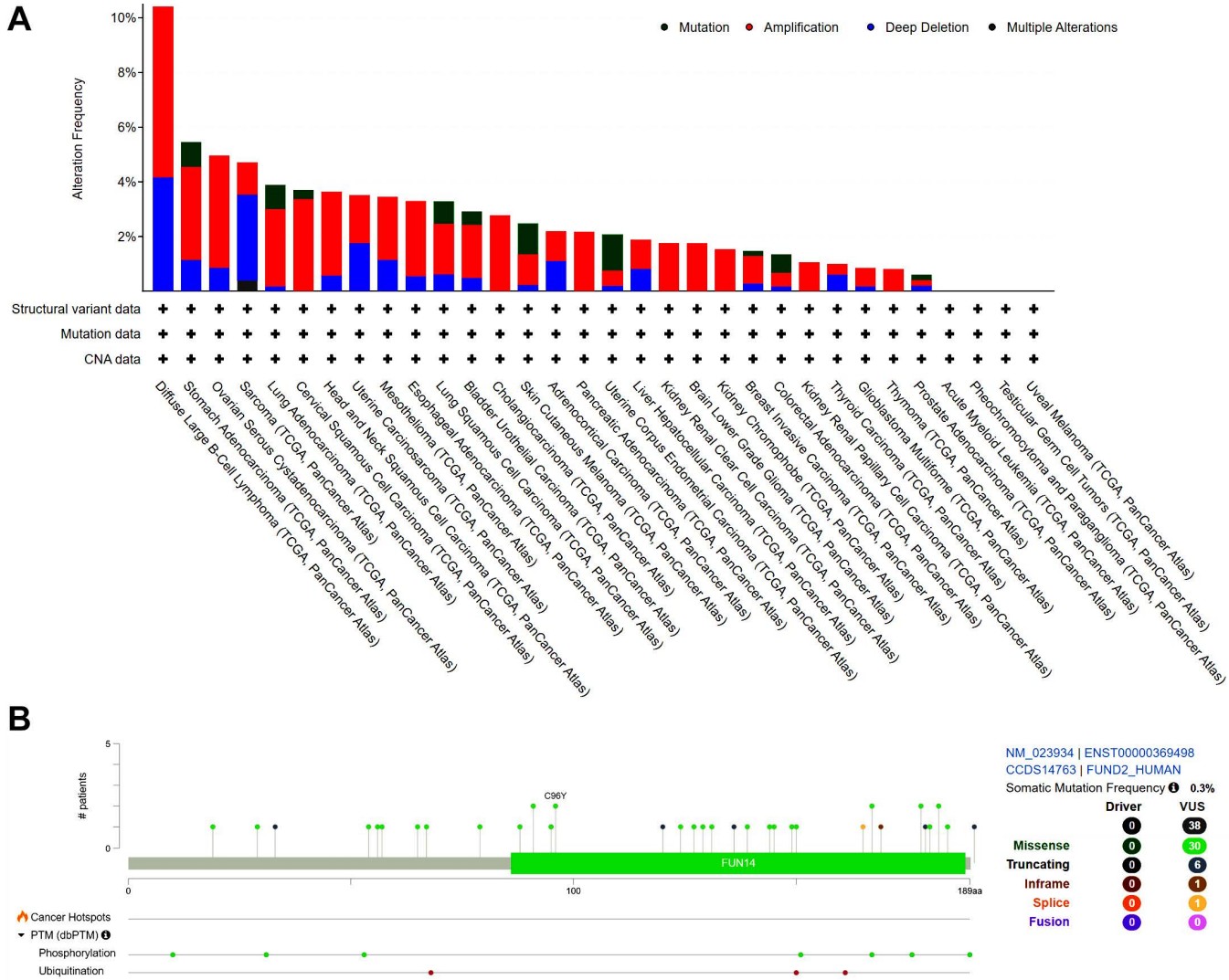

**Fig 6. Variations of FUNDC2 in pan-cancer were studied by using cBioPortal.** **(A)** Different mutations types and frequencies were summarized by cBioPortal (https://www.cbioportal.org/). Different variation types are illustrated by using different colors. **(B)** The hotspots and variations of FUNDC2 in various cancers were studied by cBioPortal. Specifically, the cancer hotspots and the phosphorylation/ubiquitination points were shown by employing different colors, respectively.

FUNDC2 expression and immune cell infiltration in cancers should be verified by experiments. Furthermore, FUNDC2 phosphorylation sites and tumor mutation hotspots should be analyzed in clinical investigations and mice models.

TIMER is characterized by analyzing immune infiltration of immune cells systematically in various cells. The statistical deconvolution method was employed in TIMER to investigate the abundance of various immune cells with cancer-infiltrating features by analyzing the expression profiles of diverse genes. A total of 10897 clinical samples across 32 different types of cancers from TCGA were included in the TIMER database, which can be used to evaluate the immune infiltration levels of various immune cell types. Additionally, there are significant differences in the levels of *FUNDC2* between different cancer tissues and the respective control tissues in most patients (Fig 1). Therefore, it is concluded that

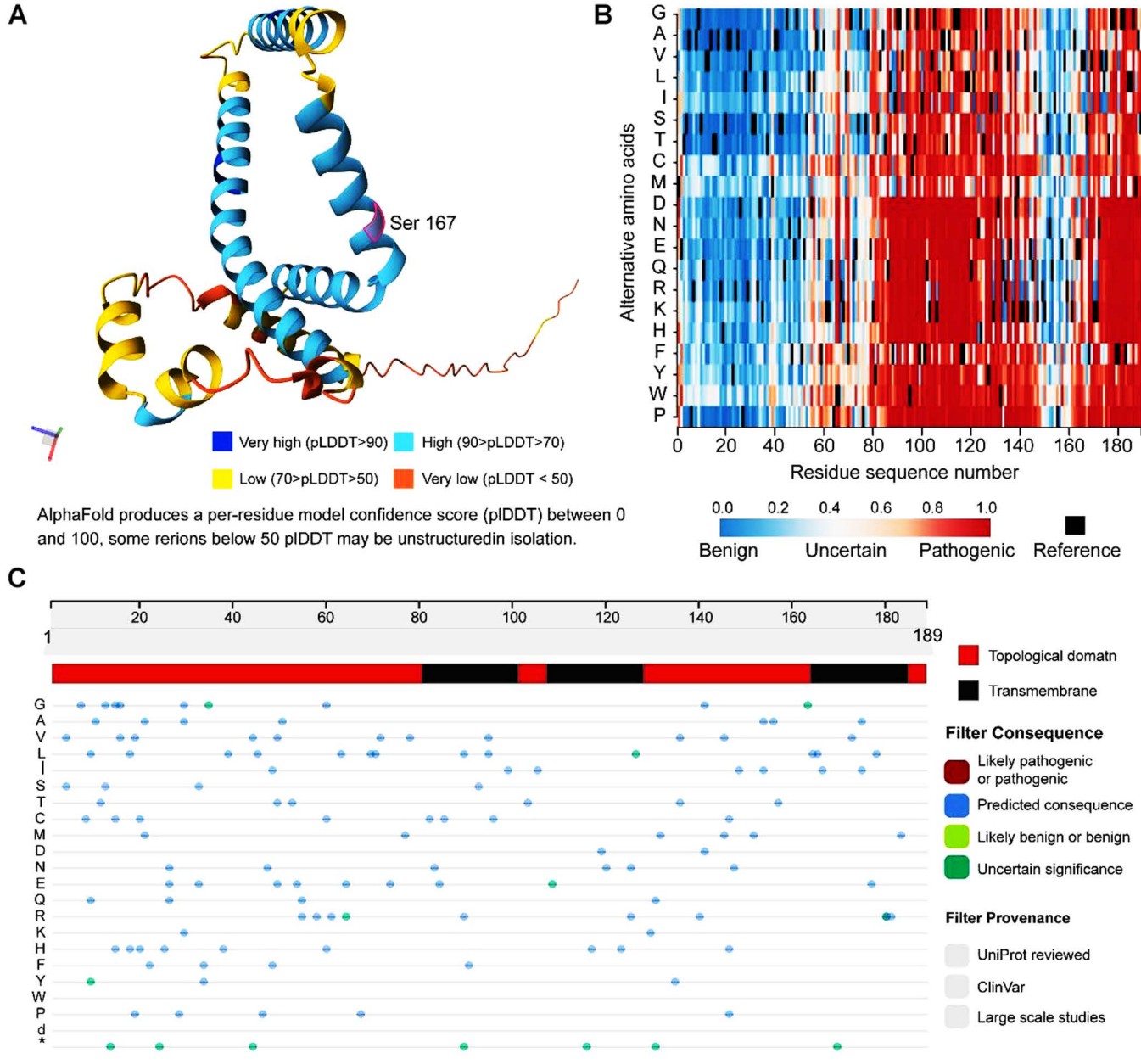

**Fig 7. Structure and pathogenicity probability of FUNDC2 were explored by AlphaFold2.** **(A)** FUNDC2 structure was predicted by AlphaFold (https://alphafold.ebi.ac.uk/). As shown, a per-residue model confidence score (pLDDT) between 0 and 100 is exhibited by AlphaFold. **(B)** Pathogenicity rate of the FUNDC2 mutation site was predicted by AlphaFold. AlphaMissense is employed to predict the pathogenicity of FUNDC2 based on Google DeepMind's AlphaFold2, which produces a score estimating the likelihood of any variant of FUNDC2 being pathogenic. **(C)** FUNDC2 domain and the pathogenicity rate of the FUNDC2 mutation site were predicted by AlphaFold.

the expression level of *FUNDC2* is highly related to the immune infiltration, and it is reasonable to deduce that the expression pattern of *FUNDC2* can affect the immune infiltration levels of the immune cells in different cancer patients.

It is well known that the immune cells in the tumor microenvironment (TME) can affect the survival time of the patients with distinct cancers, and the analysis results of the bioinformatics in the current study showed that the expression levels of

*FUNDC2* are associated with the survival of multifarious cancer patients, indicating a potential prognostic effect of *FUNDC2* in pan-cancer (Fig 2). In addition, the results in this investigation demonstrate that the levels of FUNDC2 are associated with various immune infiltration levels of B cells, DC, et al. in diverse cancers including BLCA and BRCA (Fig 3), further supporting the specific correlation of the expression of FUNDC2 with the immune infiltration of certain immune cells.

The mechanisms of why the expression levels of FUNDC2 affect the immune infiltration and the survival time of the patients with diverse cancers are studied. Early studies have showed that the immune cells resist tumors [13]. Specifically, the immune cells of the immune system especially macrophages and T cells are activated during the early stage of carcinogenesis, which kill the cancer cells to impede cancer development. However, the immune cells in the TME will favor tumor cells and enhance cancer progression when the cancers have progressed to this early stage. As known, FUNDC2 is a novel mitochondrial protein and it plays key roles in both platelet apoptosis and ferroptosis [1–3], which is implicated in immunity and inflammation. Interestingly, it has been proved that mitochondrial protein can participate in antigen presentation. The data in this study (Fig 3) show clearly that the immune infiltration of the antigen-presenting cells especially the B cells is significantly associated with the expression levels of FUNDC2 in diverse cancers. However, the mechanism of FUNDC2 in antigen presentation remains to be explored. Anyhow, all the mechanisms of FUNDC2 under the context of cancers mentioned above may be involved in immune infiltration.

It has been reported that the expression level of FUNDC2 is highly involved in TNBC and hepatocarcinoma by regulating cancer cell proliferation and invasion [6–8]. Furthermore, the results in Fig 1 illustrate clearly that the levels of *FUNDC2* in the metastatic cancer tissues (n=368) are significantly higher than those of the corresponding cancer tissues without metastasis (n=103) in SKCM. In addition, the data in Fig 3 exhibit that the level of *FUNDC2* is significantly correlated to the immune infiltration of the B cells in the tumor microenvironment. These data collectively indicate that the expression levels of *FUNDC2* may affect immune evasion and tumor microenvironment.

Importantly, numerous studies have demonstrated that the immune cells resist tumors [13]. Specifically, the immune cells especially macrophages and T cells in the tumor microenvironment are activated at the early stage of cancers to kill the cancer cells and impede cancer development. With cancer progression, the immune cells in the TME will help tumor cells to escape immune system, which is regulated by the expression of various mitochondrial proteins especially the FUN14-domain-containing proteins, suggesting that FUNDC2 may influence immune evasion and the tumor microenvironment.

The prediction of phosphorylation sites and mutations of FUNDC2 is valuable. The reliability of these predictions should be confirmed and discussed. First, the PhosphoNet is an important and reliable online tool to predict the potential phosphorylation sites of any proteins, by which numerous phosphorylation sites of various proteins have been confirmed by experimental data. Specifically, the sites of S151, T176 and S189 in FUNDC2 are also validated by experiments (Fig 5), suggesting that the predictions of phosphorylation sites of FUNDC2 are reliable. Second, diverse mutations of FUNDC2 including S167L, C96Y, and M139I are found in different cancer patients (Fig 6), as evidenced in the TCGA databases, suggesting that these variations of *FUNDC2* at least are supported by clinical data. Third, phosphorylation modifications play key roles in gene expression and regulation, and dysregulation of FUNDC2 involving phosphorylation incurs pathological phenotypes in both mice models and clinical cases [6–8]. The data from the AlphaFold online tool and Fig 7 in this study strongly show that both the predicted phosphorylation sites and mutations in FUNDC2 display high pathogenicity, indicating the reliability of the prediction data.

However, the reliability of these predicted phosphorylation sites and mutations of FUNDC2 should be further confirmed in cell lines, mice models and patients. Anyhow, the predictions of phosphorylation and mutations on FUNDC2 provide potential candidates for experiments. Additionally, whether the mutations of FUNDC2 especially on the predicted phosphorylation sites are a cause or just a consequence in cancer remains to be explored.

The clinical implications of the FUN14-domain-containing proteins [51–53] especially FUNDC2 as a biomarker are important. First, the results in Fig 2 exhibit that the expression level of FUNDC2 is significantly associated with the

survival and prognosis of patients with diverse cancers, indicating that FUNDC2 may serve as a prognostic biomarker of cancer patients. Second, the data in Fig 1 demonstrated that there are significant differences in FUNDC2 expression levels between cancer tissues and normal tissues, suggesting that changed expression pattern of FUNDC2 is helpful in the diagnosis of potential cancers. Third, various variations of FUNDC2 (Fig 6) are discovered in cancer patients, showing its potential diagnosis and treatment values. Fourth, the data in Fig 7 demonstrate the pathogenicity of every amino acid of FUNDC2, indicating its treatment implications by targeting the specific sites of FUNDC2. Anyhow, it remains to be explored by experiments and clinical studies.

Possible kinases have been predicted by employing the PhosphoNet online tool, as shown in the S1 Fig. In addition, the phosphatases responsible for FUNDC2 may be PGAM5, PP1 or PP2. Both FUNDC2 and PGAM5 are mitochondrial proteins, so it is reasonable to deduce that PGAM5 may be a potential phosphatase for FUNDC2. However, further studies should be done to validate the hypothesis.

Future studies on FUNDC2 should include clinical analysis of FUNDC2 variations in patients with cancers and mice models. In conclusion, FUNDC2 may serve as a possible prognostic biomarker in pan-cancer and the mechanism may be involved in immune infiltration.

## Supporting information

**S1 Fig. The potential protein kinases responsible for FUNDC2 predicted by PhosphoNet**. The PhosphoNet (http://www.phosphonet.ca/) is employed to predict the possible protein kinases responsible for FUNDC2 at the specific site of Ser10 (TSAPRAG**S**QVVATTA). The data show that ataxia telangiectasia and Rad3-related protein (ATR, ATR serine/threonine kinase) is most likely the protein kinase for FUNDC2 at Ser10.
(TIF)

## Author contributions

**Conceptualization:** Xirong Qiu.

**Data curation:** Shuyu Wang.

**Project administration:** Chenlu Li.

**Supervision:** Yinan Wang.

**Writing – original draft:** Xirong Qiu.

**Writing – review & editing:** Xirong Qiu.

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
