## [Decision Letter · Decision Letter 0]

13 Dec 2024

PONE-D-24-52291Expression and immunological role of FUNDC2 in pan-cancerPLOS ONE

Dear Dr. Qiu,

Thank you for submitting your manuscript to PLOS ONE. After careful consideration, we feel that it has merit but does not fully meet PLOS ONE’s publication criteria as it currently stands. Therefore, we invite you to submit a revised version of the manuscript that addresses the points raised during the review process.

We look forward to receiving your revised manuscript.

Kind regards,

Jinhui Liu

Academic Editor

PLOS ONE

Additional Editor Comments:

Authors should revise according to the suggestions of reviewers. The modifications should be marked. A point to point response letter is needed.

Reviewers' comments:

Reviewer's Responses to Questions

**Comments to the Author**

1. Is the manuscript technically sound, and do the data support the conclusions?

Reviewer #1: Yes

Reviewer #2: Yes

2. Has the statistical analysis been performed appropriately and rigorously? 

Reviewer #1: Yes

Reviewer #2: Yes

3. Have the authors made all data underlying the findings in their manuscript fully available?

Reviewer #1: Yes

Reviewer #2: Yes

4. Is the manuscript presented in an intelligible fashion and written in standard English?

Reviewer #1: Yes

Reviewer #2: Yes

5. Review Comments to the Author

Reviewer #1: The manuscript offers valuable insights into the role of FUNDC2 in cancer and presents a well-rounded analysis using several bioinformatics databases. The findings support the potential of FUNDC2 as a prognostic biomarker and provide useful information about its involvement in cancer progression and immune infiltration. However, the manuscript presents several issues, including:

1.The results are well-organized, but some figures (e.g., Figure 3 showing immune infiltration) could be discussed in more detail. Exploring how FUNDC2 expression affects immune cell infiltration in different cancer types would provide deeper insights into its role in tumor immunology.

2. The discussion is generally sound but could benefit from a more in-depth exploration of the molecular mechanisms by which FUNDC2 influences immune evasion and the tumor microenvironment. This will strengthen the conclusions drawn from the study.

3. The study acknowledges several limitations, but they could be further elaborated. For example, while the prediction of phosphorylation sites and mutations is valuable, the reliability of these predictions could be questioned. A more detailed discussion of potential biases in the data would improve transparency.

4. Expand the discussion on the clinical implications of FUNDC2 as a biomarker. Discuss how these findings might inform clinical strategies for cancer diagnosis, prognosis, and treatment.

5. While the study uses bioinformatics tools effectively, experimental validation of some of the findings, particularly related to mutations and phosphorylation sites, would greatly strengthen the manuscript.

6. The manuscript is generally well-written, but some sentences are lengthy and could be more concise. Simplifying the language will improve readability.

7. Some of the figures could benefit from clearer labeling and more detailed captions, making them easier for readers to interpret.

Reviewer #2: Review Comments:

The manuscript entitled “Expression and immunological role of FUNDC2 in pan-cancer” by Qiu, et al. presents an interesting investigation into the expression and immunological role of FUNDC2 in pan-cancer. The combined application of various bioinformatics tools, including TIMER 2.0, GEPIA, STRING, cBioPortal, PhosphoNet, and AlphaFold, to analyze the expression patterns, immune infiltration, prognosis, potential interacting proteins, phosphorylation sites, and structural predictions of FUNDC2 in pan-cancer is commendable.

The data indicate that FUNDC2 expression levels differ significantly between cancer tissues and controls, with higher levels observed in 8 types of cancers, suggesting a potential role for FUNDC2 in cancer. Furthermore, the authors found that patients with higher levels of FUNDC2 generally had longer survival times across multiple cancer types, which is very interesting. The results also demonstrates a significant correlation between FUNDC2 expression and immune infiltration of B cells in cancer patients, hinting at a possible immunological role for FUNDC2. The STRING analysis reveals potential protein interactions between FUNDC2 and other proteins, such as FUNDC1, which may be involved in cancer-related pathways. Additionally, the study predicts phosphorylation sites on FUNDC2, including a mutation site at S167, which may affect its function and interaction with other proteins. The use of AlphaFold to predict the structure of FUNDC2 adds another layer of complexity to our understanding of this protein.

Overall, this study is well-designed and carried out, providing valuable insights into the expression patterns, immunological role, and potential mechanisms of FUNDC2 in pan-cancer. The data collectively indicate that FUNDC2 is closely associated with cancers, and it may serve as a potential biomark of pan-cancer.

Many years ago, I and my collaborators first identified the FUN14-containing protein 1/2 (FUNDC1/FUNDC2), and we are trying to uncover its clinical significance but obtains little. Personally, this study by Qiu, et al. provides valuable information on role and mechanism of FUNDC2 in diseases. I strongly recommend that the manuscirpt should be accepted for publication after revision.

However, if the authors can address the following concerns, it will be better.

1. The study relies on bioinformatics tools to analyze publicly available data, but there is little experimental data. Therefore, the authors should discuss this point in the Discussion part.

2. It is well-known for the phosphorylation modification of FUNDC1/FUNDC2 in reguation of activity and function, the possible kinases and phosphatase also should be studied or predicted.

3. Some important references are missing and should be included on FUN14-domain containing proteins.

4. A conclusion should be included in the abstract part.

5. The authors should make efforts to correct and polish the English of the whole manuscript.

6. PLOS authors have the option to publish the peer review history of their article (what does this mean? ). If published, this will include your full peer review and any attached files.

**Do you want your identity to be public for this peer review?** For information about this choice, including consent withdrawal, please see our Privacy Policy .

Reviewer #1: No

Reviewer #2: **Yes: ** Prof. Weilin Zhang. 1 National Institute of Biological Sciences, Beijing 102206, China. 2 Tsinghua Institute of Multidisciplinary Biomedical Research, Tsinghua University, Beijing 100084, China.

---

## [Author Response · Author response to Decision Letter 0]

17 Jan 2025

Dec. 19th, 2024

Academic Editor

PLOS ONE

Dear Dr. Jinhui Liu:

Thank you very much for your email of Dec. 13th, 2024, with regard to our manuscript (PONE-D-24-52291) together with the comments from the reviewers. We carefully read the comments and suggestions, and have now completed a revision of the manuscript following the suggestions from the reviewers and you. We responded point by point to each reviewer’s comments as listed below. We thank the reviewers for their constructive suggestions that have improved the quality and clarity of the manuscript and we hope that the revised manuscript is acceptable for publication.

Thank you very much for your continued attention. 

Sincerely yours,

Xirong Qiu

Xirong Qiu

School of Medicine,

Lijiang Culture and Tourism College,

Lijiang, 674199, Yunnan, China.

TEL: 0086-15887878421

FAX: 0086-15887878421

E-mail: ynkmq6688@outlook.com

Our point-by-point response to the referees’ comments:

Response to Reviewer 1:

Reviewer #1: The manuscript offers valuable insights into the role of FUNDC2 in cancer and presents a well-rounded analysis using several bioinformatics databases. The findings support the potential of FUNDC2 as a prognostic biomarker and provide useful information about its involvement in cancer progression and immune infiltration. However, the manuscript presents several issues, including:

1.The results are well-organized, but some figures (e.g., Figure 3 showing immune infiltration) could be discussed in more detail. Exploring how FUNDC2 expression affects immune cell infiltration in different cancer types would provide deeper insights into its role in tumor immunology.

Response: Thank you very much for your valuable comments. We have discussed the immune infiltration in more detail including Figure 3, et al. The details were included in the “Materials and methods” part of the Revised Manuscript with Track Changes (page 4, line 109-114).

Furthermore, how FUNDC2 expression affects immune cell infiltration in different cancers were also explored to provide deeper insights into the role and mechanisms of FUNDC2 in tumor immunology in the “Discussion” part of the Revised Manuscript with Track Changes (pages 9-10, line 254-290).

TIMER is characterized with analyzing immune infiltration of immune cells systematically in various cells. The statistical deconvolution method was employed in TIMER to investigate the abundance of various immune cells with cancer-infiltrating features by analyzing the expression profiles of diverse genes. A total of 10897 clinical samples across 32 different types of cancers from TCGA were included in the TIMER database, which can be used to evaluate the immune infiltration levels of various immune cell types. Additionally, there are significant differences in the levels of FUNDC2 between different cancer tissues and the respective control tissues in most patients (Figure 1). Therefore, it is concluded that the expression level of FUNDC2 is highly related to the immune infiltration, and it is reasonable to deduce that the expression pattern of FUNDC2 can affect the immune infiltration levels of the immune cells in different cancer patients.

It is well known that the immune cells in the tumor microenvironment (TME) can affect the survival time of the patients with distinct cancers, and the analysis results of the bioinformatics in the current study show that the expression levels of FUNDC2 are associated with the survival of multifarious cancer patients, indicating a potential prognostic effect of FUNDC2 in pan-cancer (Figure 2). In addition, the results in this investigation demonstrate that the levels of FUNDC2 are associated with various immune infiltration levels of B cells, DC, et al. in diverse cancers including BLCA and BRCA (Figure 3), further supporting the specific correlation of the expression of FUNDC2 with the immune infiltration of certain immune cells.

The mechanisms of why the expression levels of FUNDC2 affect the immune infiltration and the survival time of the patients with diverse cancers are studied. Early studies have showed that the immune cells resist tumors (Yuan Q, Sun N, Zheng J, Wang Y, Yan X, Mai W, et al. Prognostic and immunological role of FUN14 domain containing 1 in pan-cancer: friend or foe? Front Oncol. 2020;9:1502. doi: 10.3389/fonc.2019.01502. PMID: 31998650.). Specifically, the immune cells of the immune system especially macrophages and T cells are activated during the early stage of carcinogenesis, which kill the cancer cells to impede cancer development. However, the immune cells in the TME will favor tumor cells and enhance cancer progression when the cancers have progressed to this early stage. As known, FUNDC2 is a novel mitochondrial protein and it play key roles in both platelet apoptosis and ferroptosis (Ma Q, Zhu C, Zhang W, Ta N, Zhang R, Liu L, et al. Mitochondrial PIP3-binding protein FUNDC2 supports platelet survival via AKT signaling pathway. Cell Death & Differentiation. 2018;26(2):321-31. Na Ta, Chuanren Qu, Hao Wu, Di Zhang, Tiantian Sun, Yanjun Li, Jun Wang, Xiaohui Wang, Tieshan Tang, Quan Chen, Lei Liu. Mitochondrial outer membrane protein FUNDC2 promotes ferroptosis and contributes to doxorubicin-induced cardiomyopathy.Proc Natl Acad Sci U S A. 2022 Sep 6;119(36):e2117396119.), which is implicated in immunity and inflammation. Interestingly, it has been proved that mitochondrial protein can participate in antigen presentation. The data in this study (Figure 3) show clearly that the immune infiltration of the antigen presenting cells especially the B cells is significantly associated with the expression levels of FUNDC2 in diverse cancers. However, the mechanism of FUNDC2 in antigen presentation remains to be explored. Anyhow, all the mechanisms of FUNDC2 under the context of cancers mentioned above may be involved in immune infiltration.

2. The discussion is generally sound but could benefit from a more in-depth exploration of the molecular mechanisms by which FUNDC2 influences immune evasion and the tumor microenvironment. This will strengthen the conclusions drawn from the study.

Response: The molecular mechanisms by which FUNDC2 influences immune evasion and the tumor microenvironment are discussed in the revised manuscript (Pages 10-11, lines 291-307).

It has been reported that the expression level of FUNDC2 is highly involved in TNBC and hepatocarcinoma by regulating cancer cell proliferation and invasion (Yin L, Cao R, Liu Z, Luo G, Li Y, Zhou X, et al. FUNDC2, a mitochondrial outer membrane protein, mediates triple-negative breast cancer progression via the AKT/GSK3β/GLI1 pathway. Acta Biochim Biophys Sin. 2023;55(11):1770-83. doi: 10.3724/abbs.2023142. PMID: 37700593. Li S, Han S, Zhang Q, Zhu Y, Zhang H, Wang J, et al. FUNDC2 promotes liver tumorigenesis by inhibiting MFN1-mediated mitochondrial fusion. Nat Commun. 2022;13(1):3486. doi: 10.1038/s41467-022-31187-6. PMID: 35710796. Delgado AP, Hamid, S, Brandao, P, & Narayanan, R. A novel transmembrane glycoprotein cancer biomarker present in the X chromosome. Cancer genomics & proteomics. 2014;11(2):81-92. PMID: 24709545.). Furthermore, the results in Figure 1 illustrate clearly that the levels of FUNDC2 in the metastatic cancer tissues (n=368) are significantly higher than those of the corresponding cancer tissues without metastasis (n=103) in SKCM. In addition, the data in Figure 3 exhibit that the level of FUNDC2 is significantly correlated to the immune infiltration of the B cells in the tumor microenvironment. These data collectively indicate that the expressions level of FUNDC2 may affect immune evasion and tumor microenvironment.

Importantly, numerous studies have demonstrated that the immune cells resist tumors (Yuan Q, Sun N, Zheng J, Wang Y, Yan X, Mai W, et al. Prognostic and immunological role of FUN14 domain containing 1 in pan-cancer: friend or foe? Front Oncol. 2020;9:1502. doi: 10.3389/fonc.2019.01502. PMID: 31998650.). Specifically, the immune cells especially macrophages and T cells in the tumor microenvironment are activated at the early stage of cancers to kill the cancer cells and impede cancer development. With cancer progression, the immune cells in the TME will help tumor cells to escape immune system, which is regulated by the expression of various mitochondrial proteins especially the FUN14-domain-containing proteins, suggesting that FUNDC2 may influence immune evasion and the tumor microenvironment.

3. The study acknowledges several limitations, but they could be further elaborated. For example, while the prediction of phosphorylation sites and mutations is valuable, the reliability of these predictions could be questioned. A more detailed discussion of potential biases in the data would improve transparency.

Response: We agree with you on the point that the prediction of phosphorylation sites and mutations of FUNDC2 is valuable. The reliability of various predictions is confirmed and discussed (Pages 11, lines 308-329).

First, the phosphoNet is an important and reliable online tool to predict the potential phosphorylation sites of any proteins, by which numerous phosphorylation sites of various proteins has been confirmed by experimental data. Specifically, the sites of S151, T176 and S189 in FUNDC2 are also validated by experiments (Figure 5), suggesting that the predictions of phosphorylation sites of FUNDC2 is reliable.

Second, diverse mutations of FUNDC2 including S167, C96Y, and M139I are found in different cancer patients (Figure 6), as evidenced in the TCGA databases, suggesting that these variations of FUNDC2 at least are supported by clinical data.

Third, phosphorylation modifications play key roles in gene expression and regulation, and dysregulation of FUNDC2 involving phosphorylation incurs pathological phenotypes in both mice models and clinical cases (Yin L, Cao R, Liu Z, Luo G, Li Y, Zhou X, et al. FUNDC2, a mitochondrial outer membrane protein, mediates triple-negative breast cancer progression via the AKT/GSK3β/GLI1 pathway. Acta Biochim Biophys Sin. 2023;55(11):1770-83. doi: 10.3724/abbs.2023142. PMID: 37700593. Li S, Han S, Zhang Q, Zhu Y, Zhang H, Wang J, et al. FUNDC2 promotes liver tumorigenesis by inhibiting MFN1-mediated mitochondrial fusion. Nat Commun. 2022;13(1):3486. doi: 10.1038/s41467-022-31187-6. PMID: 35710796. Delgado AP, Hamid, S, Brandao, P, & Narayanan, R. A novel transmembrane glycoprotein cancer biomarker present in the X chromosome. Cancer genomics & proteomics. 2014;11(2):81-92. PMID: 24709545.). The data from the alphaFold online tool and Figure 7 in this study strongly show that both the predicted phosphorylaton sites and mutations in FUNDC2 display high pathogenecity, indicating the reliability of the prediction data.

However, the reliability of these predicted phosphorylation sites and mutations of FUNDC2 should be further confirmed in cell lines, mice models and patients. Anyhow, the predictions of phosphorylation and mutations on FUNDC2 provide potential candidates for experiments. Additionly, whether the mutations of FUNDC2 especially on the predicted phosphorylation sites is a cause or just a consequence in cancer remains to be explored.

4.Expand the discussion on the clinical implications of FUNDC2 as a biomarker. Discuss how these findings might inform clinical strategies for cancer diagnosis, prognosis, and treatment.

Response: The clinical implications of FUNDC2 as a biomarker are discussed (page 12, line 330-342). First, the results in Figure 2 exhibit that the expression level of FUNDC2 is significantly associated with the survival and prognosis of patients with diverse cancers, indicating that FUNDC2 may serve as a prognostic biomarker of cancer patients. Second, the data in Figure 1 demonstrated that there are significant differences in FUNDC2 expression levels between cancer tissues and normal tissues, suggesting that changed expression pattern of FUNDC2 is helpful in the dianosis of potential cancers. Third, various variations of FUNDC2 (Figure 6) are discovered in cancer patients, showing its potential diagnosis and treatment values. Fourth, the data in Figure 7 demonstrate the pathogenecity of every amino acid of FUNDC2, indicating its treatment implications by targeting the specific sites of FUNDC2. It is suggested that the people carring variations of FUNDC2 should pay special attention to their health. Anyhow, it remains to be explored by experiments and clinical studies.

5. While the study uses bioinformatics tools effectively, experimental validation of some of the findings, particularly related to mutations and phosphorylation sites, would greatly strengthen the manuscript.

Response: Thank you very much for your suggestion. We totally agree with you on the point that experimental validation of some of the findings especially the mutations and phosphorylation sites in this study is important, which is discussed as one of the limitations. Because the authors just graduated several months ago, and there is no grants and equipments for us to do experiments, so the related experiments will be done later. We sincerely hope that you can understand us. Thank you very much! The data by bioinformatics will provide valuable information for potential choices of validation involving FUNDC2.

The reliability of various predictions is confirmed and discussed (Pages 11, lines 308-329).

First, the phosphoNet is an important and reliable online tool to predict the potential phosphorylation sites of any proteins, by which numerous phosphorylation sites of various proteins has been confirmed by experimental data. Specifically, the sites of S151, T176 and S189 in FUNDC2 are also validated by experiments (Figure 5), suggesting that the predictions of phosphorylation sites of FUNDC2 is reliable.

Second, diverse mutations of FUNDC2 including S167, C96Y, and M139I are found in different cancer patients (Figure 6), as evidenced in the TCGA databases, suggesting that these variations of FUNDC2 at least are supported by clinical data.

Third, phosphorylation modifications play key roles in gene expression and regulation, and dysregulation of FUNDC2 involving phosphorylation incurs pathological phenotypes in both mice models and clinical cases (Yin L, Cao R, Liu Z, Luo G, Li Y, Zhou X, et al. FUNDC2, a mitochondrial outer membrane protein, mediates triple-negative breast cancer progression via the AKT/GSK3β/GLI1 pathway. Acta Biochim Biophys Sin. 2023;55(11):1770-83. doi: 10.3724/abbs.2023142. PMID: 37700593. Li S, Han S, Zhang Q, Zhu Y, Zhang H, Wang J, et al. FUNDC2 promotes liver tumorigenesis by inhibiting MFN1-mediated mitochondrial fusion. Nat Commun. 2022;13(1):3486. doi: 10.1038/s41467-022-31187-6. PMID: 35710796. Delgado AP, Hamid, S, Brandao, P, & Narayanan, R. A novel transmembrane glycoprotein cancer biomarker present in the X chromosome. Cancer genomics & proteomics. 2014;11(2):81-92. PMID: 24709545.). The data from the alphaFold online tool and Figure 7 in this study strongly show that both the predicted phosphorylaton sites and mutations in FUNDC2 display high pathogenecity, indicating the reliability of the prediction data.

However, the reliability of these predicted phosphorylation sites and mutations of FUNDC2 should be further confirmed in cell lines, mice models and patients. Anyhow, the predictions of phosphorylation and mutations on FUNDC2 provide potential candidates for experiments. Additionly, whether the mutations of FUNDC2 especially on the predicted phosphorylation sites is a cause or just a consequence in cancer remains to be explored.

6. The manuscript is generally well-written, but some sentences are lengthy and could be more concise. Simplifying the language will improve readability.

Response: Thank you very much for your expert comments and helpful suggestions. We have tried our best to improve the English of the manuscript. Specifically, shorter sentences are used, which is more concise now. In addition, the language has been simplified to improve readability. Thank you for your advices.

7. Some of the figures could benefit from clearer labeling and more detailed captions, making them easier for readers to interpret.

Response: Clearer labeling and more detailed captions are

---

## [Decision Letter · Decision Letter 1]

31 Jan 2025

Expression and immunological role of FUNDC2 in pan-cancer

PONE-D-24-52291R1

Dear Dr. Qiu,

We’re pleased to inform you that your manuscript has been judged scientifically suitable for publication and will be formally accepted for publication once it meets all outstanding technical requirements.

Kind regards,

Jinhui Liu

Academic Editor

PLOS ONE

Additional Editor Comments (optional):

The authors have addressed the reviewers' concerns properly and revised the manuscript accordingly. The manuscript can be accepted for publication in its current form

Reviewers' comments:

Reviewer's Responses to Questions

**Comments to the Author**

1. If the authors have adequately addressed your comments raised in a previous round of review and you feel that this manuscript is now acceptable for publication, you may indicate that here to bypass the “Comments to the Author” section, enter your conflict of interest statement in the “Confidential to Editor” section, and submit your "Accept" recommendation.

Reviewer #1: All comments have been addressed

Reviewer #2: All comments have been addressed

2. Is the manuscript technically sound, and do the data support the conclusions?

Reviewer #1: Yes

Reviewer #2: (No Response)

3. Has the statistical analysis been performed appropriately and rigorously? 

Reviewer #1: Yes

Reviewer #2: Yes

4. Have the authors made all data underlying the findings in their manuscript fully available?

Reviewer #1: Yes

Reviewer #2: Yes

5. Is the manuscript presented in an intelligible fashion and written in standard English?

Reviewer #1: Yes

Reviewer #2: Yes

6. Review Comments to the Author

Reviewer #1: (No Response)

Reviewer #2: The authors have adequately addressed all the concerns, so I strongly recommend that it can be accepted for publication ASAP. Thank you.

7. PLOS authors have the option to publish the peer review history of their article (what does this mean? ). If published, this will include your full peer review and any attached files.

**Do you want your identity to be public for this peer review?** For information about this choice, including consent withdrawal, please see our Privacy Policy .

Reviewer #1: No

Reviewer #2: **Yes: ** Weilin Zhang

---

## [Editor Report · Acceptance letter]

PONE-D-24-52291R1

PLOS ONE

Dear Dr. Qiu,

I'm pleased to inform you that your manuscript has been deemed suitable for publication in PLOS ONE. Congratulations! Your manuscript is now being handed over to our production team.

Kind regards,

on behalf of

Dr. Jinhui Liu

Academic Editor

PLOS ONE